# A Low Phase Noise Frequency Synthesizer with a Fourth-Order RLC Loop Filter

Xinyu Zhang [1,2], Qifei Du [1,2,*], Cheng Liu [1], Hao Zhang [1], Yue Ma [1,2], Yefei Li [3] and Jinhuan Li [3]

1   National Space Science Center, Beijing 100190, China
2   University of Chinese Academy of Sciences, Beijing 100190, China
3   Shanghai Institute of Satellite Engineering, Shanghai 200240, China
*   Correspondence: dqf@nssc.ac.cn

**Abstract:** The current work employs the HMC830 phase-locked loop chip to design a frequency synthesizer operating in the L-band. The frequency synthesizer can provide a local oscillation signal for the RF receiver front end. This article employs the phase-locked synthesis technique to describe the design scheme. Due to the advantages of the passive loop filters, such as simplicity, low cost, and low phase noise, a passive fourth-order RLC loop filter is proposed to improve the output signal quality and reduce phase noise. The performance of this loop filter is compared with the passive fourth-order RC loop filter. The effects of these two loop filters on phase noise, loop capture time, and spur suppression are analyzed. Subsequently, the design scheme, simulation analysis, and test results of the frequency synthesizer are presented under these two loop filters. The test results indicate that the passive fourth-order RLC loop filter outperforms the passive fourth-order RC loop filter; its output signal phase noise is higher than −100 dBc/Hz@1 kHz, loop capture time is less than 100 us, and spur suppression is better than 60 dBc. This frequency synthesizer can provide high-performance local oscillation signals for wireless communication equipment such as transmitters and receivers. It meets the application requirements of many radio communication circuit structures and has good application prospects.

**Keywords:** PLL; loop filter; phase noise; frequency synthesizer

## 1. Introduction

With the growth and application of wireless communication technology, it is crucial to ensure the reliability and stability of the frequency source's output signal [1–3]. A frequency synthesizer is the core component of the RF front end in a satellite radio communication system. It can provide a precise and stable frequency as the local oscillation signal for wireless communication devices [4]. RF engineers face design difficulties such as high stability and low phase noise frequency sources. Frequency synthesis techniques include direct analog frequency synthesis, indirect frequency synthesis, and direct digital frequency synthesis [5–7]. The direct analog frequency synthesis technique is realized by multiplying, dividing, and mixing one or more base signal sources to generate many discrete frequency output signals [8]. A crystal oscillator generates the reference signal. The direct digital frequency synthesis technique employs digital sampling and storage technology to perform frequency synthesis using the phase concept. As an indirect frequency synthesis technique, the phase-locked frequency synthesis is realized by locking the frequency of the voltage-controlled oscillator (VCO) to a specific frequency through a phase-locked loop (PLL), and the required frequency is generated and output by the VCO [9,10]. Compared with the other two techniques, the indirect frequency synthesis technique can appropriately select the target frequency signal without many filters. It also has a simple structure, low power consumption, and high output signal purity, which are conducive to miniaturization and integration [11].

This article employs the phase-locked frequency synthesis approach to design a frequency synthesizer operating in the L-band. The phase-locked loop system is introduced in detail. A passive fourth-order RLC loop filter is proposed, and its performance is compared with the passive fourth-order RC loop filter in terms of phase noise, loop capture time, and spur suppression. The test results give the output frequency of 1426 MHz at the detection frequency of 50 MHz. The design provides a reference significance to the design of the L-band frequency source.

## 2. PLL Principle and Chip Introduction

### 2.1. Basic Principle of PLL

A phase-locked loop (PLL) is a phase feedback control circuit that adjusts its internal oscillator's frequency and phase using an external input reference signal [12]. It mainly comprises a phase detector (PD), a loop filter (LF), a frequency divider (FD), and a voltage-controlled oscillator (VCO) to form a closed-loop system. Figure 1 shows a typical schematic of the phase-locked loop [13].

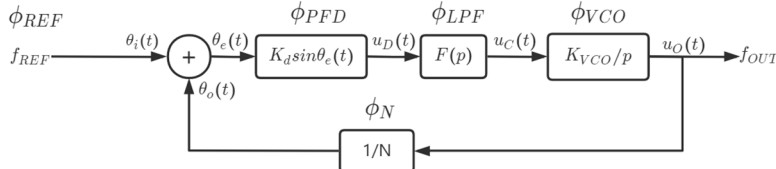

**Figure 1.** PLL schematic.

$K_{VCO}$ is the voltage control sensitivity, $K_d$ represents the voltage transfer coefficient, $F(s)$ describes the filter transfer function, and $G(s)$ represents the open-loop gain of the loop circuit. As shown in Figure 1, the PLL is a phase-negative feedback system where the input phase is $\theta_i(t)$, the VCO feedback output phase is $\theta_o(t)$, and the phase error is $\theta_e(t)$. The phase relationship is described as follows [14]:

$$\theta_e(t) = \theta_i(t) - \theta_o(t) = \theta_i(t) - K_{VCO}K_d\frac{F(p)}{pN}sin\theta_e(t) \tag{1}$$

where $p$ is the derivative operator. When $\theta_e(t)$ is small enough, $sin\theta_e(t)$ can be approximated as $\theta_e(t)$. $K_{VCO}K_d = K$ (1) can be transformed into the following complex frequency domain:

$$s\theta_e(s) = s\theta_i(s) - \frac{KF(s)\theta_e(s)}{N} \tag{2}$$

From (1), we have

$$\theta_o = K\frac{F(s)}{sN}(\theta_i - \theta_o) \tag{3}$$

From (1) and (3), the closed-loop transfer function of the loop circuit is obtained in terms of the transfer function $F(s)$ as follows [12]:

$$H(s) = \frac{\theta_o}{\theta_i} = \frac{K_{VCO}K_dF(s)}{sN + K_{VCO}K_dF(s)} = \frac{G(s)}{1 + G(s)} \tag{4}$$

where

$$G(s) = \frac{K_{VCO}K_dF(s)}{sN} \tag{5}$$

The loop filter is critical to the overall design of the frequency synthesizer. It can pass DC signals and suppress high-frequency AC signals, interference, and harmonic components. The loop filter type and bandwidth determine essential parameters of the output signal, such as phase noise, lock time, and loop stability. Although it has a simple circuit structure, it significantly influences the operation of the whole system.

### 2.2. Introduction of Loop Filter

Loop filters include the RC integral filter, the passive proportional-integral filter, and the active proportional-integral filter, as shown in Table 1. The RC integral filter is also known as the lagging filter due to its phase lag characteristics. Its output voltage is approximately proportional to the input voltage integral for sufficiently high frequencies, and the phase tends to be 90°. This type of filter has rarely been utilized in the literature. The passive proportional-integral filter maintains a fixed proportional input–output voltage relationship in the high-frequency range, with the phase asymptotic to 0°. When the phase increases, it exhibits the phase lead correction effect, improving the closed-loop stability. The proposed frequency synthesizer employs this type of filter. The active proportional-integral filter has the characteristics of a low-pass filter, and the same period phase frequency characteristics also have the role of lead correction. Generally, the higher the order of the loop filter, the better the phase performance [15].

**Table 1.** Loop filter characteristics (first-order).

| Loop Circuit Type | Noise (AMP) | Phase | Adjustable Parameter |
|---|---|---|---|
| RC integral filter | no | phase lag | 1 |
| passive proportional-integral filter | no | lead correction | 2 |
| active proportional-integral filter | yes | lead correction | 2 |

### 2.3. Introduction of the HMC830 Chip

The HMC830 is a broadband PLL chip from Hittite. The chip has an integrated voltage-controlled oscillator (VCO), a VCO output divider, a phase detector (PD), and a δ-σ modulator.

Its output frequency range is 25 MHz~3 GHz. It can work in integer or fractional frequency divider mode. It has industry-leading, ultra-low noise, and ultra-low spurious signal characteristics. The output power range of the chip is 0~9 dBm (adjustable in 3 dB steps), and the typical value of the output power is 6 dBm. The phase detection frequency inside the chip is up to 100 MHz. Increasing the phase detection frequency reduces the phase noise and shortens the loop capture time to obtain a high-quality output signal. The chip also has a δ-σ modulator for accurate frequency modes, which allows the output frequency of the chip to perform without a frequency error.

## 3. Scheme Design

The microcontroller (MCU) chip is C8051F330 from SILICON. The reference frequency source is a 50 MHz oven controlled crystal oscillator (OCXO). The reference signal is not frequency-divided, and the phase detection frequency is 50 MHz. The voltage regulator chip HMC1060LP3E provides the circuit voltage.

Figure 2 shows the system architecture. The MCU controls the PLL through its registers. The SPI protocol communicates between the MCU and the PLL. The OCXO generates the reference signal through the R divider of the PLL. The PD outputs a voltage signal by comparing the reference signal with the output signal of the VCO after the N divider. This voltage signal is linearly related to the phase difference between the two signals. The loop filter eliminates the noise and high-frequency components of the voltage signal. The filtered signal is fed into the VCO, and the VCO outputs the corresponding signal at a specific frequency. This signal is then fed back to the PD. The PD compares the feedback signal with the reference signal to make their phase difference zero. The loop circuit goes into a locked state. At this point, the VCO outputs a stable single-frequency signal, which is output via the K divider.

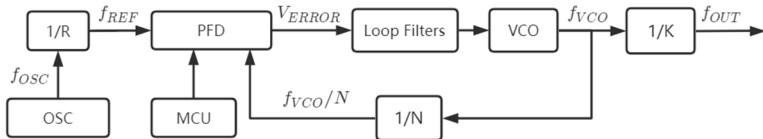

**Figure 2.** Frequency synthesizer system architecture.

The system's frequency division is described as follows [16]:

$$
\begin{cases}
f_{PD} = f_{REF} = \frac{f_{OSC}}{R} = \frac{f_{VCO}}{N} \\
f_{OUT} = \frac{f_{VCO}}{K} \\
f_{VCO} = \frac{f_{OSC}}{R}\left(N_{int} + N_{frac}\right)
\end{cases}
\tag{6}
$$

In order to alleviate the impact of phase noise on the system, a new passive fourth-order RLC filter is proposed as the loop filter. ADIsimPLL is utilized to design the two types of loop filters. ADIsimPLL is the software for the simulation design of PLL frequency synthesizers.

In the passive fourth-order RC filter, C1, C2, C3, and C4 are filter capacitors, and R1, R2, and R3 are filter resistors, forming a π-type RC filter circuit. The signal is first filtered by C1, which filters out most AC components. R1 and C2 form a passive proportional-integral filter. R2, C3, R3, and C4 form a two-stage voltage divider circuit. Due to the small capacitive reactance of C3 and C4, the voltage divider attenuates the AC component significantly, thus realizing the filtering.

The topology and components of this loop filter differ from the passive fourth-order RC filter. The passive fourth-order RLC filter comprises a pre-filter C3 and an inverted L filter (L1, C2). The filter is coupled with a series inductor to obtain a better roll-off characteristic [17]. The inductor has a self-inductance effect and high inductive resistance to AC. However, it cannot reduce the output voltage compared to the resistor. When passing current, the inductor generates an electromotive force at each terminal to suppress the current changes such that it can operate as a filter. As the current increases, some of it will be stored in the inductor, gradually increasing the current. Meanwhile, as the current decreases, the reverse electromotive force hinders its decrease. Accordingly, a smooth DC current is obtained. Let us compare these two filter circuits. The filter resistor has the same resistance to DC and AC. In contrast, the filter inductor has a large inductance to AC and a small resistance to DC. This improves the filtering effect without reducing the DC output voltage. Therefore, the RLC filter is superior to the RC filter and obtains a smoother signal. Figure 3 compares the two circuit topologies.

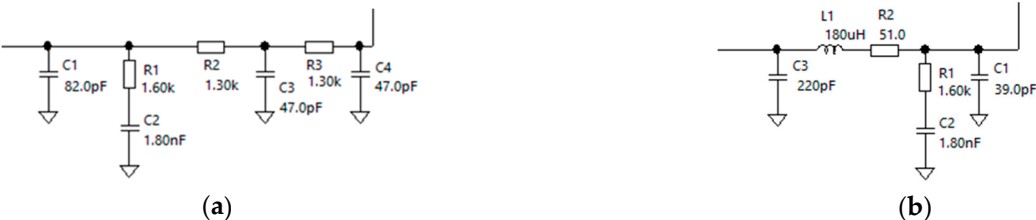

**(a)**                                                                 **(b)**

**Figure 3.** Loop structure: (**a**) passive fourth-order RC filter and (**b**) passive fourth-order RLC filter.

The closed-loop stability can be realized if, and only if, all roots of the characteristic equation of the closed-loop system have negative real parts [18]. It should be noted that stability is independent of the zero location. Table 2 shows the closed-loop system poles obtained by combining Equations (4), (7), and (8). As shown in Table 2, and since all poles of the closed-loop system are located in the left half plane, the circuit is stable.

**Table 2.** The closed-loop system poles.

|  | Passive Fourth-Order RC Filter | Passive Fourth-Order RLC Filter |
|---|---|---|
| poles | $-4.2670 \times 10^7$ $-0.6242 \times 10^7$ $-0.0348 \times 10^7$ $0$ | $(-8.1664 + 8.6381i) \times 10^6$ $(-8.1664 - 8.6381i) \times 10^6$ $(-0.3503 + 0.0000i) \times 10^6$ $0$ |

Figure 4 shows the design schematic and the circuit board.

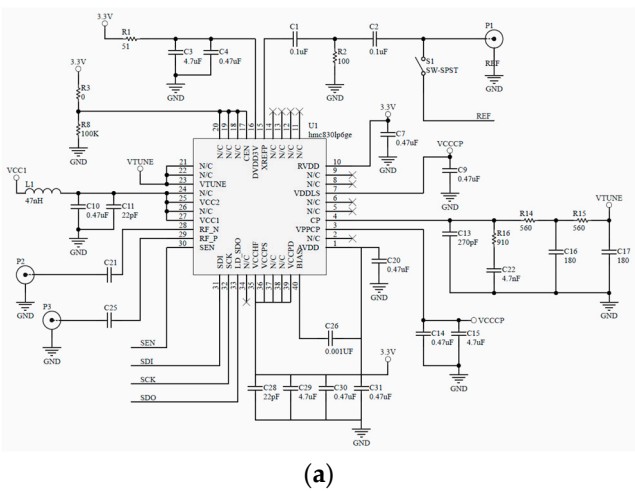

(**a**)

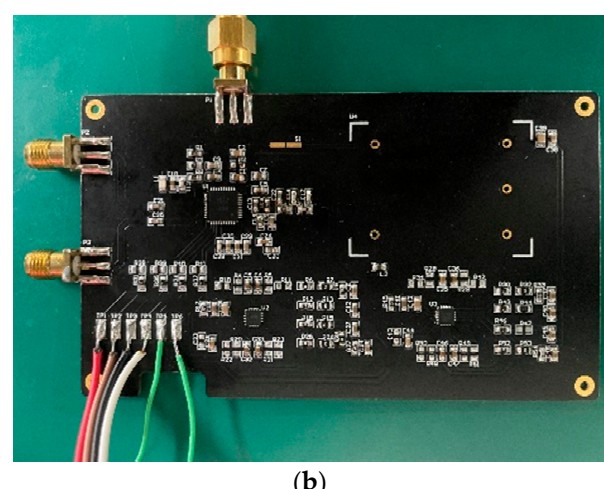

(**b**)

**Figure 4.** Circuit design: (**a**) circuit design of the PLL section and (**b**) the circuit board.

## 4. Simulation Analysis

This section simulates and analyzes the passive fourth-order RC filter and the passive fourth-order RLC filter. The filtering performance of the two filters is discussed, including the roll-off characteristic and zero-pole distribution. Phase noise and capture time are simulated for the phase-locked loop based on the two filters. The simulation results are analyzed.

### 4.1. Filter Performance Analysis

The transfer functions of the two filters and their zero-pole point distributions are as follows:

$$F_1(s) = \frac{1 + R_1C_2s}{R_3R_2C_4C_3R_1C_2s^3 + (R_2C_3R_1C_2 + R_2C_4R_1C_2 + R_3C_4R_1C_2 + R_3R_2C_4C_3)s^2 + (R_1C_2 + R_2C_3 + R_2C_4 + R_3C_4)s + 1} \quad (7)$$

$$F_2(s) = \frac{1 + R_1C_2s}{C_1R_1L_1C_2s^3 + (C_1L_1 + L_1C_2 + C_1R_1R_2C_2)s^2 + (C_1R_2 + R_2C_2 + R_1C_2)s + 1} \quad (8)$$

The filter's performance significantly depends on its zero-pole point distribution. In Table 3, the zero-pole point distributions of both filters indicate that the poles are placed in the left half-plane, demonstrating that both are stable systems. However, one pole of the passive fourth-order RC filter is closer to the imaginary axis than the passive fourth-order RLC filter. Thus, its decay time is relatively longer, leading to a slower transient response and degrading system performance [19]. The poles are determined only by the filter topology. The imaginary part of the poles corresponds to sinusoidal oscillations. The larger the value of the imaginary part, the higher the oscillation frequency, which means the more violent the oscillation. Accordingly, the superiority of the passive fourth-order RLC filter to the passive fourth-order RC filter is demonstrated.

**Table 3.** Zero-pole location distributions of the two filters.

| | Passive Fourth-Order RC Filter | Passive Fourth-Order RLC Filter |
|---|---|---|
| zeros | $-3.4722 \times 10^5$ | $-3.4722 \times 10^5$ |
| | $-4.2670 \times 10^7$ | $-8.1664 \times 10^6 + 8.6381 \times 10^6 i$ |
| poles | $-0.6242 \times 10^7$ | $-8.1664 \times 10^6 - 8.6381 \times 10^6 i$ |
| | $-0.0348 \times 10^7$ | $-0.3503 \times 10^6$ |

In practice, the filter's amplitude–frequency characteristic curve cannot reach the rectangular shape in the ideal case. Therefore, the roll-off characteristic can be defined to measure the ideal degree of a filter, determining the transition zone between the passband and the stopband and the verticality degree of this transition zone, that is, the approximation degree of the rectangular response. As shown in Figure 5, the roll-off values of the passive fourth-order RC and RLC filters are 34.62 dB/decade and 39.95 dB/decade, respectively. The roll-off characteristic of the passive fourth-order RLC filter is more significant than that of the passive fourth-order RC filter, demonstrating the better filtering effect of the former.

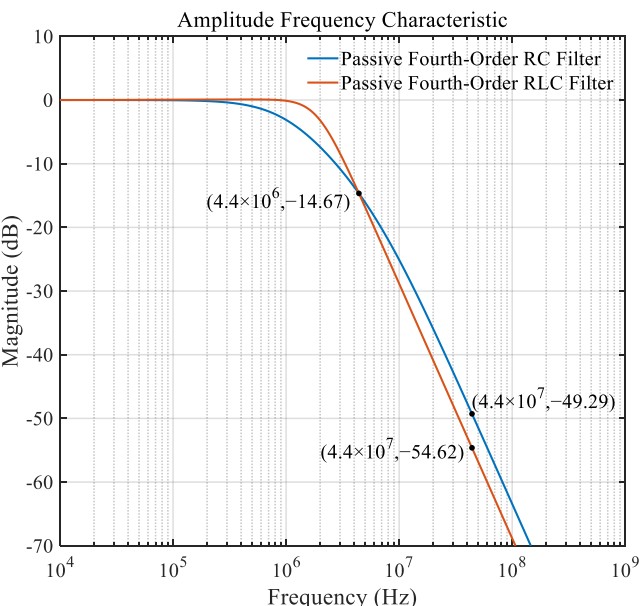

**Figure 5.** Amplitude-frequency response characteristic curve.

*4.2. Phase Noise Analysis*

Phase noise is the power per unit bandwidth ($P_{SSB}$) to the total signal power ($P_S$), deviating from the carrier frequency $f_m$ within a specific range. Its unit is dBc/Hz and can be described as follows:

$$L(f_m) = \frac{P_{SSB}}{P_S} \tag{9}$$

According to Figure 1, $\phi_{REF}$, $\phi_{PFD}$, $\phi_{LPF}$, $\phi_{VCO}$, and $\phi_N$ describe the noise introduced by the reference signal, the phase detector, the loop filter, the voltage-controlled oscillator, and the programmable N divider, respectively. Let $S_{REF}$, $S_{PFD}$, $S_{LPF}$, $S_{VCO}$, and $S_N$ denote the corresponding component's phase noise power spectral density. $S_{TOT}$ is the total phase noise of the system. The phase noise introduced by the loop filter is negligible. When the loop is locked, and since the system is linear time-invariant [20,21], the following relation can be obtained by the superposition principle:

$$S_{TOT} = \left(S_{REF}^2 + S_N^2\right) \cdot \left[\frac{G(s)}{1 + G(s)H(s)}\right]^2 + S_{PFD}^2 \cdot \left(\frac{1}{K_d}\right)^2 \cdot \left[\frac{G(s)}{1 + G(s)H(s)}\right]^2 + S_{VCO}^2 \cdot \left[\frac{1}{1 + G(s)H(s)}\right]^2 \tag{10}$$

Figure 6 shows the input noise introduced by each part of the loop and the total output phase noise.

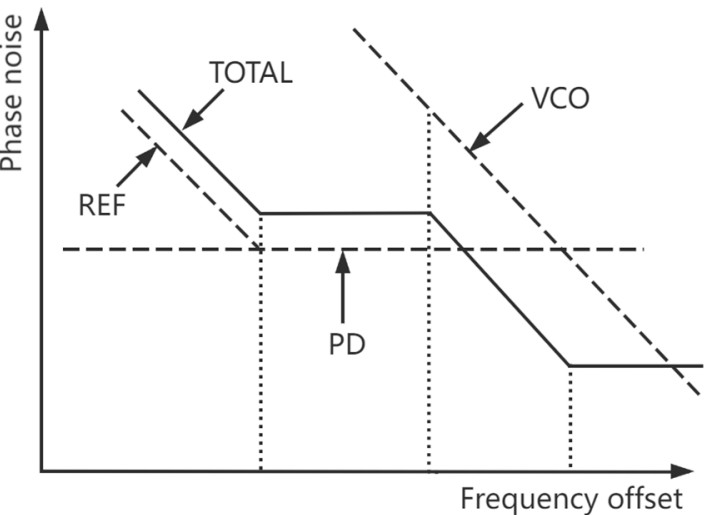

**Figure 6.** Single-loop PLL phase noise model.

The phase noise introduced by the reference source, the phase detector, and the frequency divider is low-pass. This is the primary factor influencing the total phase noise in the loop bandwidth. Moreover, the phase noise caused by the VCO is high-pass, mainly affecting the out-of-band phase noise [22]. The in-band output signal's phase noise is presented in (11), where $PN_{SYNTH}$ is the normalized noise ground of the PD, and $N$ describes the frequency division ratio.

$$PN = PN_{SYNTH} + 10\log f_{PD} + 20\log N \tag{11}$$

From (11), there are three ways to promote the system's total phase noise: increasing the phase detection frequency, reducing the frequency division ratio N, and employing a phase detector with a small noise ground. ADIsimPLL can simulate the output signal's phase noise, loop capture time, and other simulation results. The phase noise simulation results are shown in Figure 7.

Since the phase noise of the frequency synthesizer focuses on the proximal phase noise, the proximal phase noise is the phase noise in the loop bandwidth. Thus, we mainly analyze the proximal phase noise of both circuits. From Table 4, it can be seen that the proximal phase noise of the passive fourth-order RLC filter is superior to that of the passive fourth-order RC filter. In contrast, the distal phase noise, that is, the phase noise outside the loop bandwidth is inferior.

**Table 4.** Phase noise simulation results.

| Loop Circuit Type | Phase Noise (dBc/Hz) | | | |
|---|---|---|---|---|
| | 1 KHz | 10 KHz | 100 KHz | 1 MHz |
| passive fourth-order RC filter | −113.3 | −115.5 | −113.4 | −140.7 |
| passive fourth-order RLC | −113.3 | −116.0 | −113.9 | −138.4 |

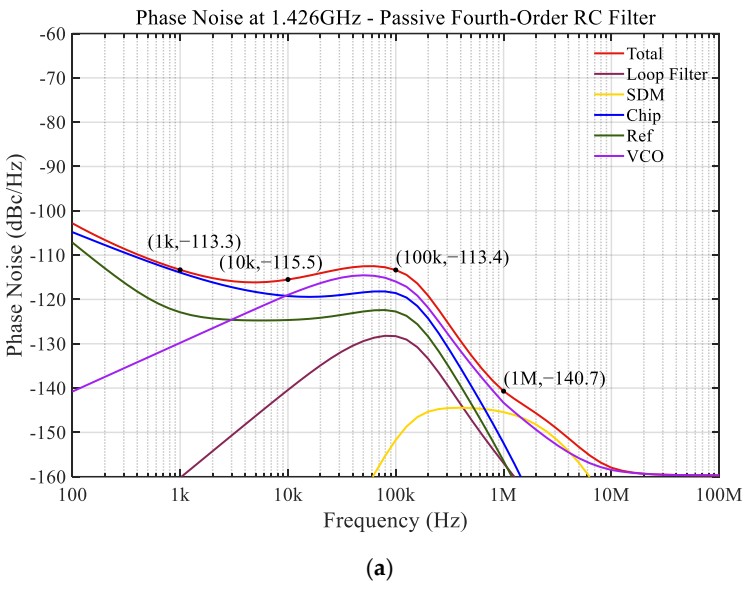

**(a)**

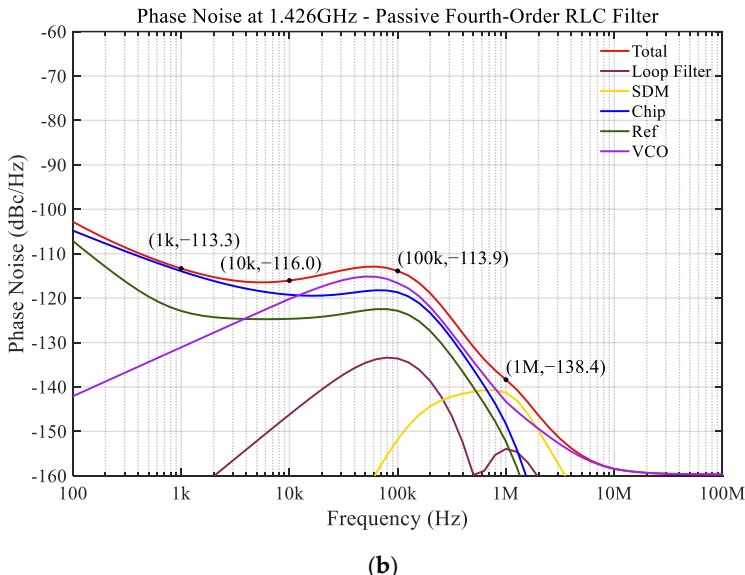

**(b)**

**Figure 7.** Phase noise simulation results: (**a**) passive fourth-order RC filter and (**b**) passive fourth-order RLC filter.

### 4.3. Loop Capture Time Analysis

Equation (12) describes the loop capture time $T_p$. It indicates when the loop reaches the locked state from the starting unlocked state.

$$T_p = \frac{\Delta\omega_0^2}{2\xi\omega_n^3} \tag{12}$$

where $\Delta\omega$ is the capture belt. It refers to the maximum initial frequency difference that can be locked through the frequency traction loop. $\omega_n$ is the bandwidth of the loop filter, and $\xi$ is the system damping coefficient. With a constant damping factor and capture band, the loop capture time is reduced by reducing the loop bandwidth. Figure 8 shows the loop capture time of the simulation.

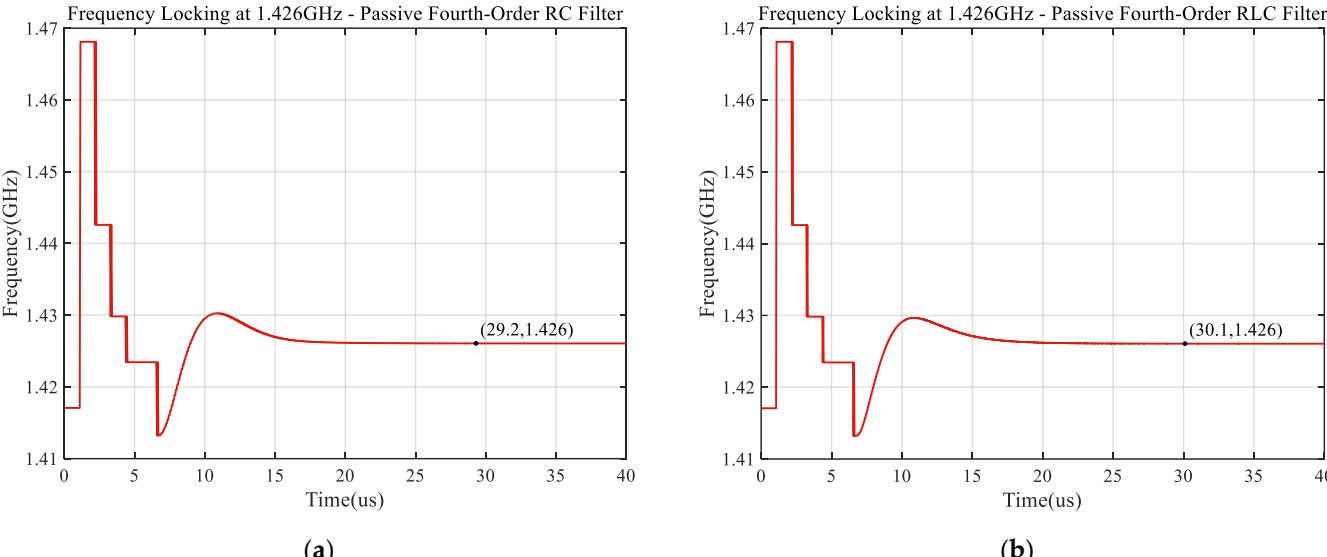

**(a)**     **(b)**

**Figure 8.** Loop capture time simulation results: (**a**) passive fourth-order RC filter and (**b**) passive fourth-order RLC filter.

As can be seen in Table 5, the capture times are the same at about 30 us. That is because both loops have the same bandwidth [23]. However, this is the ideal case of capturing time. Due to many external factors, the actual capture time can become larger.

**Table 5.** Loop capture time simulation results.

|  | Passive Fourth-Order RC Filter | Passive Fourth-Order RLC Filter |
|---|---|---|
| loop capture time (us) | 29.2 | 30.1 |

## 5. Experimental Results

This section evaluates the frequency synthesizers based on the passive fourth-order RC and RLC filters. Experiments evaluate the two circuits' phase noise, loop capture time, and spur suppression. The test results are analyzed for comparison.

### 5.1. Phase Noise Test

The instrumentation adopts the E4440A spectrometer from Keysight and the MSO64B oscilloscope from Tektronix. The passive fourth-order RC filter and the passive fourth-order RLC filter are chosen as the loop filter. The phase noise of a signal is evaluated with an output frequency of 1426 MHz.

Figure 9 presents the output signal's frequency spectrum. The output signal power is 5.36 dBm, which is lower than the typical output value of 6 dBm due to some attenuations caused by the test cable and the instrument.

The experimental phase noise is shown in Figure 10. Compared with the passive fourth-order RC filter, the passive fourth-order RLC filter has better phase noise proximal to the useful signal, and its surrounding noise is weaker. Therefore, the passive fourth-order RLC filter has a better filtering effect. The experimental results correspond to the theoretical derivation and simulation results. The output phase noise of the passive fourth-order RLC filter reaches −105.59 dBc@1 KHz. Due to the superposition of external noise, such as circuit and instrumentation, the experimental result deviates from the simulation result (below 8 dB). In addition, there is a curve bump at the distal of the signal due to the high-pass noise from the chip's internal VCO.

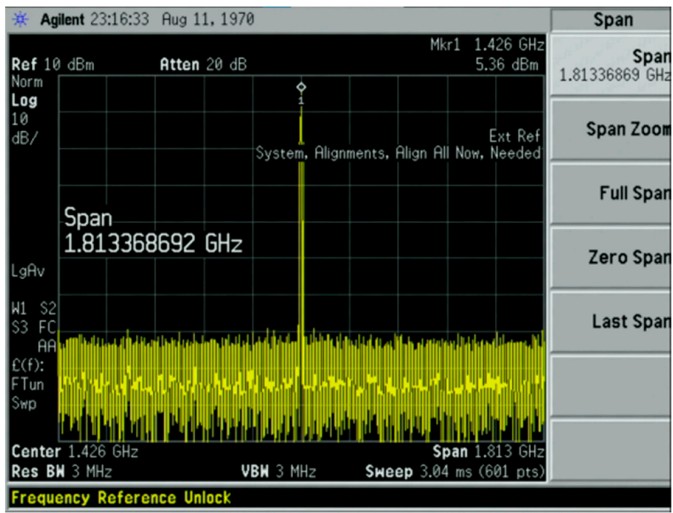

**Figure 9.** The output signal's frequency spectrum.

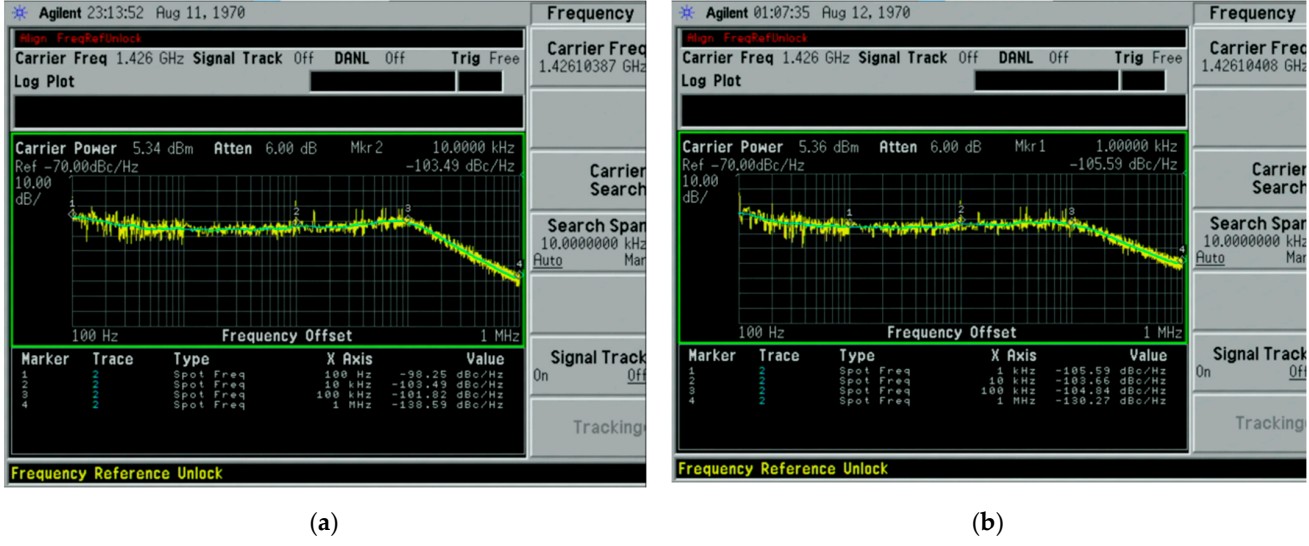

| (**a**) | (**b**) |

**Figure 10.** Output signal phase noise: (**a**) passive fourth-order RC filter and (**b**) passive fourth-order RLC filter.

The phase noise of the designed frequency synthesizer can reach −93.84 dBc/Hz@1 KHz, −82.56 dBc/Hz@1 KHz, 87.27 dBc/Hz@1 KHz, and −86.5 dBc/Hz@1 KHz using an active second-order filter in [24], an active third-order filter in [25], a passive third-order filter in [11], and a third-order microstrip filter in [26], respectively. As shown in Table 6, and since the phase noise of the frequency synthesizer in this design is superior to the mentioned references, this design has low phase noise performance.

**Table 6.** Comparison of phase noises of different filters.

| Loop Circuit Type | Phase Noise (dBc/Hz) | | | |
|---|---|---|---|---|
| | 1 KHz | 10 KHz | 100 KHz | 1 MHz |
| passive fourth-order RC filter | −98.25 | −103.49 | −101.82 | −138.59 |
| passive fourth-order RLC filter | −105.59 | −103.66 | −104.84 | −130.27 |
| active second-order filter [24] | −93.84 | −95.23 | −92.07 | \ |
| active third-order filter [25] | −82.56 | −93.34 | −99.77 | −96.79 |
| passive third-order filter [11] | −87.27 | −90.27 | −93.26 | −112.97 |
| third-order microstrip filter [26] | −86.5 | −85 | −87 | −115 |

### 5.2. Loop Capture Time Test

The loop capture times of the loop output signal of both loop filters are evaluated with an output frequency of 1426 MHz. From Table 7, it can be seen that the capture times of the two filter circuits are equal, about 94 us. This is because the filter types are similar, and the loop bandwidths are the same at 160 kHz. The loop capture time indicates when the loop reaches the locked state from the starting unlocked state. The tested trigger level is the level falling edge trigger after completely writing the MCU program to the PLL.

**Table 7.** Comparison of loop capture times of different filters.

| Loop Circuit Type | Loop Capture Time |
|---|---|
| passive fourth-order RC filter | 93.69 us |
| passive fourth-order RLC filter | 94.02 us |
| active second-order filter [24] | 2.5 ms |
| active second-order filter [27] | 14.4 ms |

The loop capture time of the designed frequency synthesizer can reach 2.5 ms and 14.4 ms using active second-order filters in [24,27], respectively. As shown in Table 7, and since the loop capture time of the frequency synthesizer in this design is superior to the mentioned references, this design has better loop capture performance.

### 5.3. Spur Suppression Test

Spurs are unwanted periodic components of the spectrum. Spurs can reduce the signal-to-noise ratio of a wireless communication system and increase signal jitter [28]. For both the passive fourth-order RC and RLC filters, the spur suppression of a signal is evaluated with an output frequency of 1426 MHz. The experimental spur suppression results are shown in Figure 11.

Since the output frequency is not an integer multiple of the reference frequency, the fractional-N frequency synthesizer necessarily outputs spurious signals [28]. The division ratio modulated by the frequency divider causes the quantization noise. The spectrum of its quantization error causes noise and spuriousness in the corresponding position of its spectrum. In addition to the above spuriousness, reference frequencies appear as spurious signals in the spectrum. As shown in Figure 11, the spurious signals are mainly at 50 MHz, 100 MHz, and 150 MHz. From the experimental data, the passive fourth-order RLC filter is superior to the passive fourth-order RC filter in spur suppression. The spurious signals are all frequency interferences from the reference source. Spur suppression is nearly 70 dB and 90 dB for the proximal and distal phase noise, respectively.

The spur suppression of the designed frequency synthesizer is more significant than 47.04 dBc/Hz in [25], 58 dBc/Hz in [11], and 45 dBc/Hz in [26]. As shown in Table 8, the spur suppression of the frequency synthesizer in this design is superior to the mentioned references. Therefore, this frequency synthesizer provides good suppression of spurious signals.

**Table 8.** Comparison of spur suppressions of different filters.

| Loop Circuit Type | Spur Suppression (dBc/Hz) | | |
|---|---|---|---|
| | 50 MHz | 100 MHz | 150 MHz |
| passive fourth-order RC filter | −69.48 | −73.35 | −85.92 |
| passive fourth-order RLC | −69.80 | −73.89 | −87.12 |
| active second-order filter [25] | | <−47.04 | |
| passive third-order filter [11] | | <−58 | |
| third-order microstrip filter [26] | | <−45 | |

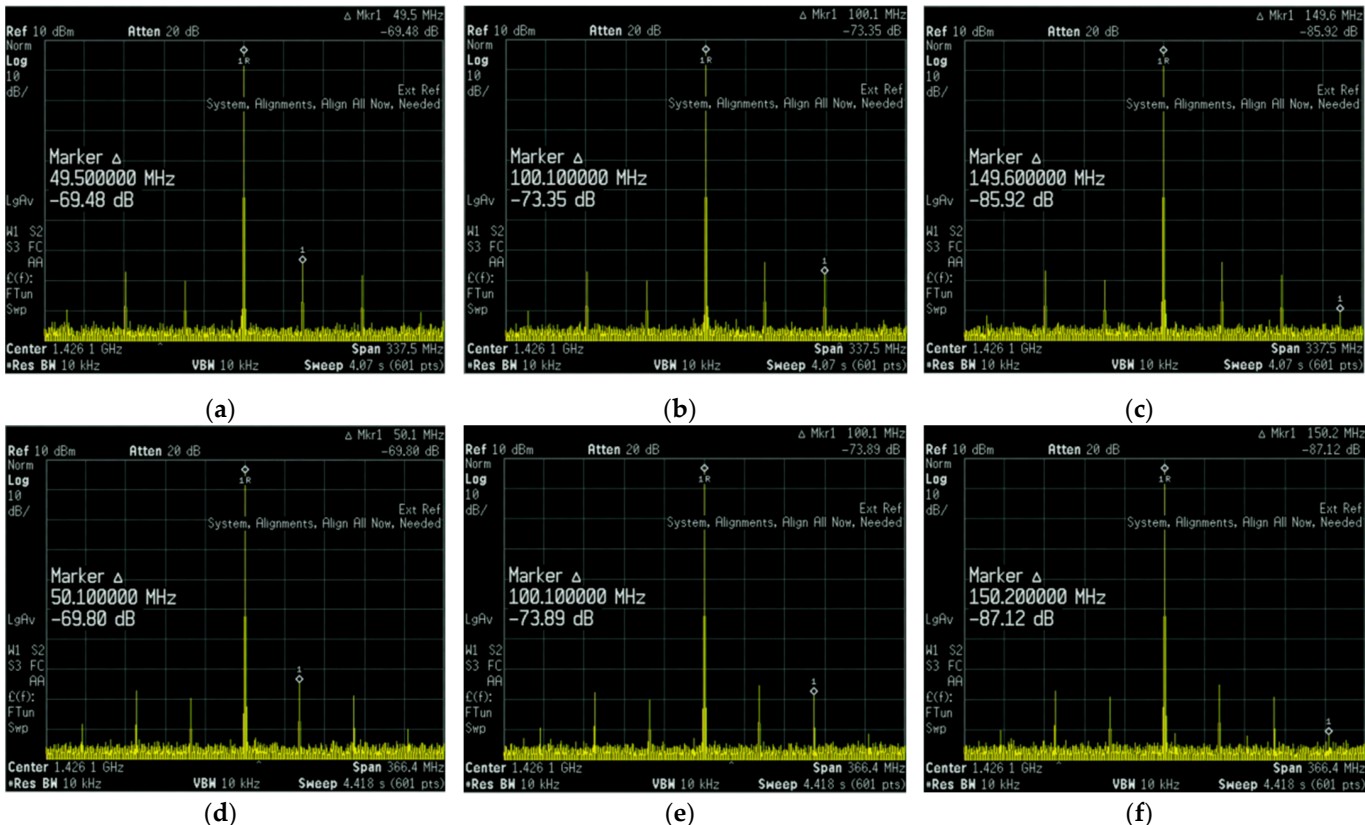

**Figure 11.** Spur suppression: (**a**–**c**) passive fourth-order RC filter and (**d**–**f**) passive fourth-order RLC filter.

## 6. Experimental Conclusions

This article designed a frequency synthesizer with low phase noise and low spurious signals in the L-band using the HMC830 chip. The SPI protocol communicates between the MCU and the PLL. The voltage conversion chip provides the voltage to the PLL chip. The phase noise of the test output signal can reach −105.59 dBc@1 KHz. The proximity spur suppression is up to 70 dB. The loop lock time is as short as 94 us. This frequency synthesizer provides a high-quality output signal, which can be employed as the local oscillation signal source of the RF receiving front end, fully meeting the wireless communication requirements. Due to the importance of the loop filter in the performance of the frequency synthesizer, different filter types with different loop bandwidths can affect the output signal quality. Thus, a passive fourth-order RLC loop filter is proposed to improve output signal quality. This loop filter is compared with the passive fourth-order RC loop filter. By analyzing the transfer functions of the two filters and the zero-pole distribution, observing the simulation and experimental results, and analyzing the output signal parameters, such as phase noise, capture time, and spur suppression, it can be concluded that the output signal quality of the frequency synthesizer of the passive fourth-order RLC filter is better than that of the passive fourth-order RC loop filter.

**Author Contributions:** Conceptualization, X.Z., Q.D., Y.M. and J.L.; methodology, X.Z. and Q.D.; software, X.Z.; validation, X.Z., C.L., H.Z. and Y.L.; investigation, X.Z., Q.D. and C.L.; writing—original draft preparation, X.Z. and Q.D.; writing—review and editing, X.Z., Q.D., C.L., H.Z., Y.M., Y.L. and J.L. All authors have read and agreed to the published version of the manuscript.

**Funding:** This research was funded by the National Natural Science Foundation of China (grant no. 42074042 and no. 42104032) and the Youth Cross Team Scientific Research Project of the Chinese Academy of Sciences (grant no. JCTD-2021-10).

**Data Availability Statement:** https://www.scidb.cn/en/s/fqEVze.

**Acknowledgments:** Qifei Du, Cheng Liu, and Zhihui Lv are gratefully acknowledged for their fruitful comments and helpful discussions.

**Conflicts of Interest:** The authors declare no conflict of interest.

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
