# Peer review of "A Low Phase Noise Frequency Synthesizer with a Fourth-Order RLC Loop Filter"

_electronics, doi:10.3390/electronics12010224_

Round 1
Reviewer 1 Report
1. Referencing is not a proper format
2. Practical results can be better interpreted
3. How can be verified the stability/reliability of the designed hardware?
Reviewer 2 Report
The authors presented a study on a low phase noise frequency synthesizer with a fourth-order 2 RLC loop filter. The work is quite interesting and will help the research community by providing state-of-the-art information. However, there are some flaws with manuscript that should be revised carefully.
Throughout the manuscript there many typos and grammatical issues. Here are some of them:
In line 28, "signal[1-4]." must be replaced with "signal [1-4]."
In line 32, "source" must be replaced with "sources"
In line 38, "which is conducive" must be replaced with "which are conducive"
The equations (1-6) if taken from some reference then should be cited properly.
The method reported in Ref 6 - 12 should be explain briefly in introduction to strength the literature review and increase the interest of reader.
The the two circuit topologies shown in Fig. 3 are of author own work. If not, then must be properly cited to avoid any copyright issue. Moreover, it is better to explain briefly the difference between both circuits.
All figure should be resized to a same dimensions, unnecessary big size picture degrade the presentation of the work.
The quality of Fig. 6, 8 and 9 should be improved to increase the readability of the data.
The claim by authors "This paper designs a frequency synthesizer with low phase noise and low spurious in L-band using the HMC830 chip" need a comparison table, otherwise this statement is just a theory.
References should be formatted properly as per journal format.
Ref. 2, 7, 12, 17, 22, 23 are outdated and should be replaced with some more recent work to empower the literature section.
Reviewer 3 Report
This manuscript presents the advantages of using passive fourth-order RLC loop filter over RC filter. Both theory and experiments are discussed. Overall quality is good. Some parts of the English needs to be improved.
Reviewer 4 Report
In this work, the authors developed and characterized a frequency synthesizer operating in the L-band using the commercial HMC830 phase-locked loop chip. The characteristics of this device are quite interesting from the point of view of applications. The manuscript is clear and the conclusions are in accordance with results, therefore, it can be considered for publication if the following revision is considered:
- Please avoid repeating “paper” several times in the abstract and in the whole manuscript.
- Indicate in the abstract possible applications of this filter.
- Please compare the characteristics of this filter with other published in literature. A table systematizing the characteristics of this device with those published is welcome.
- Please avoid the use of formula word to indicate equation. Verify it in whole the manuscript.
- If possible, indicate the error bars in the values listed in the tables.
- Please increase the font size in the figures.
Round 2
Reviewer 2 Report
The authors carefully addressed the raised concern. Thus, the manuscript is recommended for publication.